# Influence of Thermal Processing on In Vitro Starch Digestibility in Cereal-Based Infant Foods [note 1]

**DOI:** 10.3390/foods14081367

**Published:** 2025-04-16

**Authors:** Marianela D. Rodríguez, Nicolás F. Bongianino, Alberto E. León, Mariela C. Bustos

**Affiliations:** 1Instituto de Ciencia y tecnología de los Alimentos Córdoba (ICyTAC), CONICET—Universidad Nacional de Córdoba (UNC), Córdoba CP 5000, Argentina; marianelarodriguez@agro.unc.edu.ar (M.D.R.); nicolasbongianino@agro.unc.edu.ar (N.F.B.); mbustos@agro.unc.edu.ar (M.C.B.); 2Facultad de Ciencias Agropecuarias, Universidad Nacional de Córdoba (UNC), Córdoba CP 5000, Argentina

**Keywords:** infant in vitro digestion, complementary feeding, starch hydrolysis, nutrient bioaccessibility

## Abstract

Early-life nutrition is crucial for healthy infant development. This study explored the effects of high-temperature (30 min, 121 °C) and high-humidity treatments on the starch properties and digestibility of infant purees made from wheat, rice, and maize. Purees were prepared using whole grains (WGs), whole grain flours (WGFs), and flour suspensions (FSs) subjected to thermal treatment. Untreated whole grain samples from each cereal served as controls. Samples were analyzed for microstructure, thermal properties, viscosity, and starch digestibility. Microstructural analysis revealed partial to complete loss of amyloplast birefringence, particularly in FS. The thermal treatment reduced peak viscosity in WGs, WGFs, and FSs. Also, the flour suspensions showed lower thermal stability and a phytic acid content reduction of 30%. In vitro digestion revealed a significant reduction in total hydrolyzed starch (THS) in wheat- (27.8 g/100 g starch) and maize- (11.3 g/100 g starch) WG purees compared to controls. In contrast, WGF purees showed significant increases in THS: 29% (wheat), 70% (rice), and 92% (maize). FS purees also showed significant increases in THS (57.4, 39.3, and 45.4 g/100 g starch for wheat, rice, and maize, respectively), alongside a decrease in resistant starch. In conclusion, thermal treatment modulates starch digestibility and viscosity properties in a cereal-dependent manner, offering a potential approach to optimize infant nutrition.

## 1. Introduction

Nutrition in the early years of life is crucial for the healthy development of children [1]. Complementary feeding, which involves introducing foods to supplement a milk-based diet, typically begins around 6 months of age and extends until 24 months, when breastfeeding usually ends [2]. Most of the weaning foods are semisolid with a soft texture. These foods are traditionally composed of staple cereals, roots, and legumes prepared either individually or as a mixture in a purée [3]. 

Thermal sterilization of infant foods is a common technology that is supposed to have little potential for further development. Optimum thermal sterilization of food always requires a compromise between the beneficial and destructive effects of heat on the food [4]. Autoclaving, heat-moisture treatment (HMT), and annealing are optional thermal treatments that utilize controlled heat and moisture to modify food components, primarily starch [5]. Autoclaving is distinguished by high temperatures (>100 °C) and low pressure for sterilization and enzymatic inactivation. HMT employs lower temperatures (80–140 °C) and restricted humidity (<35%). Annealing involves high moisture content (≥60–65%) at temperatures between the glass transition temperature (Tg) and the gelatinization onset temperature (To) of starch, requiring precise control of temperature and time. At an industrial level, autoclaving is advantageous because it allows for easy control of temperature, time, and humidity in the established equipment, which supports standardization and consistency in large volumes. This differs from the precise humidity control required in HMT or the temperature range control needed in annealing [6]. The autoclaving conditions chosen in the study can be replicated using a pressure cooker at home, demonstrating the practical applicability of the findings. Additionally, autoclaving is often used for microbiological safety, which is an important factor for baby food. Some criteria for selecting the optimal sterilization method are food safety-affecting factors: product range, nutritional influence, sensory quality, and packaging. Although autoclaving can cause structural and nutritional defects and consumes a lot of energy, the autoclaving process generates a direct transport of energy to the product (steam), ensuring the inactivation of microbes and bacterial spores [7,8,9]. The effects of thermal treatments on cereals are highly variable, influenced by factors such as cereal type, temperature, time, and humidity during processing. In the context of complementary feeding foods, some articles applied standard sterilization conditions (121 °C for 30–37 min), which also produced a reduction in phytic acid content [10,11,12]. So, the effect of autoclaving on the physicochemical properties of starch [13] is well established; however, as far as we know, there is little information about how the digestibility properties of infant food are affected, considering the gastrointestinal tract conditions of babies.

It is important to highlight that designing infant foods is not always a straightforward process [1,14]. Whereas some underweight babies may need food with high levels of rapidly digestible starch to provide quick energy without taxing their metabolism, overweight babies may benefit from high levels of slowly digestible starch to promote satiety [15]. Since infant foods should match diverse energy requirements, it is important to evaluate the effects of thermal treatments on key aspects, such as starch digestibility and anti-nutritional factors [16].

In vitro digestion under infant conditions is a technique to assess the effects of thermal treatments on the starch digestibility of infant cereal-based food. It can equally be effective in evaluating the presence of antinutritional factors (e.g., phytates) that typically interfere with the absorption of iron, zinc, and calcium, which are essential during the complementary feeding period to ensure proper child development [17,18].

Although numerous studies have examined the effects of thermal treatments like autoclaving on isolated starches [13,18,19], many articles focus on isolated starch or flour, and autoclaving for above 1 h; as far as we know, no evaluation of conventional autoclaving (121° C for 30 min) on whole grains and flours or flour suspensions intended for infant food has been evaluated. The application of autoclaving could be an interesting way to obtain healthy infant food that is easy to cook at home and safe. Moreover, it allows for the study of the effect on the entire matrix with the presence of other nutrients and structures, providing a more comprehensive view of the potential modifications and nutritional benefits. Furthermore, this study examined the effects of autoclaving, a widely used sterilization technique for complementary infant foods, on starch properties in durum wheat, brown rice, and white maize. Autoclave thermal treatment (121 °C for 30 min) alters starch digestibility in cereal-based infant foods, varying by cereal type and presentation (whole grains, flours, flour suspensions). This could help optimize complementary food nutrition for infants with diverse dietary needs.

## 2. Materials and Methods

### 2.1. Materials

Wheat (*Triticum durum* sp.), brown short-grain rice (*Oryza sativa* subsp. *japonica*), and white maize (*Zea mays* L.) were purchased from a local market and stored at 4 °C. Porcine pancreatic α-amylase (A3176), porcine pancreatic pepsin (P7000), pancreatin (P7545), and bile salts (B8756) were purchased from Sigma-Aldrich (Buenos Aires, Argentina).

### 2.2. Methods

#### 2.2.1. Preparation and Autoclaving of Cereal Samples

Three sample types were prepared from each selected cereal: wheat, rice, and maize. These included whole grains (WGs), whole grain flour (WGFs) obtained by milling the grains using a Cyclotec CT193 cyclonic mill (Foss, Suzhou, China), 0.5 mm silver—200 g of each were placed in 1 L beakers and autoclaved—and a whole-flour/water suspension (FS) prepared by mixing 100 g with 500 mL of water in 1 L beakers (1:5 ratio). Autoclaving was performed at 121 °C for 30 min at 0.11 MPa. Following autoclaving, FS samples were freeze-dried to remove excess water. Untreated whole grains from each cereal served as controls: wheat (W), rice (R), and maize (M).

#### 2.2.2. Polarized and Light Microscopy

The treated samples and controls, which had been milled beforehand, were mounted onto microscope slides for observation. They were examined using a Leica DM5000B microscope (Leica Microsystems, Wetzlar, Germany), both with and without polarized light. Images were captured using a Leica DFC450 camera at 20× magnification (Wetzlar, Germany), which was connected to a PC equipped with InFocus software (×64, 4.12.25159.20240404).

#### 2.2.3. Rapid Visco-Analysis (RVA)

A rapid viscosity analyzer (RVA-4500, Perten Instruments, Macquarie Park, Australia) was used to measure the pasting properties of samples using the standard pasting method. Samples (3.5 g dry basis (db.)) were gently suspended in 28 mL of distilled water, and the RVA was operated according to standard method 1 (short-time procedure) from the RVA manual. Peak viscosity (PV), breakdown (BD), final viscosity (FV), setback (SB), and peak viscosity time (tPV) were calculated. Controls and WG samples were milled before analysis. All measurements were performed in duplicates, and controls and WG samples were milled before analysis.

#### 2.2.4. Differential Scanning Calorimetry (DSC)

The gelatinization characteristics of autoclaved and control samples were measured using a differential scanning calorimeter (DSC 823, Mettler-Toledo, Greifensee, Switzerland), with thermograms being evaluated by STARe software (V 9.00, Mettler Toledo, Greifensee, Switzerland). Control and autoclaved samples were weighed (5 mg, db.) in aluminum pans with 10 µL of deionized water and hermetically sealed, allowed to equilibrate for 24 h before analysis at room temperature [20]. After equilibration, samples were heated at 10 °C/min from 20 °C to 120 °C to observe the presence of any residual enthalpy gelatinization peak. Thermal transitions were characterized through the peak temperature (Tp), and endothermic enthalpy (ΔH, expressed in J/g of starch in the sample) and the temperature range were calculated as the difference between the conclusion and onset temperature. All measurements were performed in duplicates, and controls and WG samples were milled before analysis.

#### 2.2.5. Infant Food Preparation

Control and WG samples (200 g) were cooked in boiling water (800 mL) until they reached a soft texture, allowing them to be easily mashed [21]. Cooking times were 30 min for wheat and maize controls, as well as their respective WG samples: 45 min for rice control and WG rice. WGFs and FSs, having been ground prior to autoclaving, were suspended in water at a 1:5 ratio and cooked for 20 min at 80 °C until the mixture became viscous. All cooking times were determined by visually checking for full gelatinization between two glass slides, as commonly done for pasta samples, in addition to the specified textural quality of cooked samples. This method is similar to traditional home cooking practices, indicating its practical application.

#### 2.2.6. Phytic Acid Content in Infant Food

Infant food prepared from the control and autoclaved samples was freeze-dried and analyzed for the phytic acid content throughout the determination of phosphorus content according to Navarro [22]. Dodecasodium phytate was used for calibration, and results were expressed as mg of phosphorus per g of phytic acid.

#### 2.2.7. In Vitro Digestion of Infant Food

In vitro digestion of the purees was carried out by simulating the digestive conditions of 6–12-month-old infants, according to the method described by Rodriguez [21]. Briefly, to simulate digestion, fluids were prepared as follows:

Simulated salivary fluid (SSF): 15.1 mM potassium chloride, 3.7 mM potassium dihydrogen phosphate, 13.6 mM sodium bicarbonate, 0.15 mM magnesium chloride hexahydrate, 1.5 mM calcium chloride hydrate, and 75 U/mL of porcine pancreas amylase; the pH was adjusted to 7.0.

Simulated gastric fluid (SGF): 13 mM potassium chloride, 94 mM sodium chloride, and 485 U/mL of porcine pancreas pepsin; the pH was adjusted to 5.3.

Simulated intestinal fluid (SIF): 10 mM potassium chloride, 85 mM sodium bicarbonate, 164 mM sodium chloride, 3 mM calcium chloride hydrate, 0.2 mg/mL of porcine pancreas pancreatin (8×USP), and 2.5 mg/mL of bile salts; the pH was adjusted to 6.6.

The model utilized a 50/50 (*w*/*v*) ratio of food to SSF, a 63:37 ratio for oral content to SGF, and a 62:38 ratio for gastric content to SIF. Five grams of each puree infant food were employed.

Starch hydrolysis was monitored by analyzing the reducing sugars at the different stages of in vitro digestion, using the 3,5-dinitrosalicylic acid (DNS) method. Starch hydrolysis fractions were calculated for oral, gastric, and intestinal phases. Total hydrolyzed starch (THS) was calculated as the amount of starch digested at the end of the digestion in vitro. Resistant starch (RS) was determined as the difference between the total starch in the sample and the total hydrolyzed starch.

#### 2.2.8. Statistical Analysis

Data were analyzed using InfoStat software (InfoStat, Cordoba, Argentina, Version 2022). An analysis of variance (ANOVA) was performed, with a significance level of 0.05, using the DGC comparison test. Each infant food sample was prepared twice and then analyzed in duplicate.

## 3. Results and Discussion

### 3.1. Optical Microscopy

Microscopic images of control and autoclaved cereal samples taken under light field and polarized light are shown in Figure 1 and Figure 2, respectively. Morphological differences were observed in starch granules of each cereal; that is, rice starch granules were agglomerated and the smallest compared to wheat and maize (Figure 1). Additionally, fragments of amyloplast with embedded starch were present in WG, WGF, and control samples (Figure 1). In WG and WGF samples, a heterogeneous population of starch granules was revealed, including intact, deformed, and empty granules (Figure 1). In addition, deformed granules exhibited a partial loss of birefringence, while empty granules displayed a complete loss, indicating modifications in their crystalline structure (Figure 2), particularly in WGF cereals. By contrast, FS samples showed fragments of empty amyloplast (Figure 1) and a complete absence of starch granules, as starch gelatinization could take place during autoclaving, as evidenced by the total loss of birefringence (Figure 2).

### 3.2. Pasting Properties of Samples

Autoclaved starches from different sources have been studied in recent years, and the effects on physicochemical properties have been established, revealing that amorphous and crystalline regions undergo structural changes, which result in alterations in the granular swelling, functional, thermal properties, and susceptibility towards enzymes and acids [13,23]. In this sense, results on pasting properties showed a decrease in the viscosity values (Figure 3), probably due to the disrupted starch granules and partial solubilization caused by temperature, which pointed out the limited development of the starch granules, as some authors reported [23,24]. Autoclaved whole grains showed a decrease throughout the entire viscosity curve, exhibiting a reduction in breakdown (starch resistance to disintegration) and setback (starch retrogradation tendency) (Figure 3). The peak viscosity of WG wheat and WG rice samples was lower than their respective controls, decreasing from 1540 to 469 cP for wheat and 2731 to 1421 cP for rice. In addition, for wheat, WG samples also showed a significant reduction in the final viscosity (1196 cP) compared to the control (2346 cP), while no significant changes were observed in WG rice, WG maize, and WG cereals (Figure 3). The pasting properties changes could be attributed to a new surface layer formed during the heat treatment, which restricts the penetration of water in the granules and delays and reduces their swelling, as reported by others [25].

Similar results were found when whole-grain flours were autoclaved. All samples presented a decrease in peak viscosity in comparison to the control: 289 cP in WGF wheat, 1386 cP in WGF rice, and 547 cP in WGF maize, denoting a more pronounced effect compared to WG samples (Figure 3), as observed in micrographs where partial loss of birefringence was observed (Figure 2). These results indicate that when the WGF was autoclaved, the starch is more exposed to temperature and humidity, which could favor structural reorganization, reducing its ability to absorb water and swell [26]. A significant decrease in final viscosity was observed in WGF samples, which may be attributed to alterations in amylose retrogradation and reduced amylose leaching under limited water conditions, as reported by previous studies [7,25].

Finally, due to FS samples undergoing starch gelatinization, a significant decrease in peak viscosity (730 cP for FS wheat, 2465 cP for FS rice, and 641 cP for FS maize) was observed, as well as in BD, SB, and FV (Figure 3). In this sense, rice was less affected, probably due to its small-sized granules (Figure 1).

The results highlight that not only the morphology of the starch granule, which varies according to its botanical origin, but also the type of sample treated, namely, grains, flours, or suspensions, influence the intensity and/or effects of autoclaving, as expected [7,27,28].

### 3.3. Thermal Properties of Samples

The thermal properties of control and autoclaved cereals are summarized in Table 1. The peak (Tp) transition temperatures did not show significant differences between the WG and controls; only wheat showed some variation in WGF samples. However, for FS samples, the Tp was lower than controls (Table 1), as expected due to the lower molecular organization and a complete alteration in the initial starch structure as observed in micrographs with complete loss of birefringence (Figure 2). The range of transition temperatures exhibited minimal variation in WG and WGF samples except in maize, which suggests that autoclaving led to a reduction in the degree of crystallite heterogeneity in the last [29].

Thermal enthalpy (∆H) values of autoclaved cereals significantly (*p* ≤ 0.05) varied as compared to those of the controls (Table 1). These results emphasized that this might be due to different levels of organization in granular (control) and autoclaved samples that need to be elucidated with further studies, as suggested by others [24].

WG samples showed different changes in the transition enthalpy compared to control samples (*p* ≤ 0.05) (Table 1). In contrast, in WGF samples, a significant increase in ΔH was observed compared to their respective controls (Table 1), consistent with findings reported by Dundar [24]. The results of FS samples showed the greatest decrease in ΔH, evidencing a less ordered structure than the untreated samples [30].

### 3.4. Phytic Acid Content in Infant Purees

The phytic acid content of the control samples highlights that maize presented the lowest value (0.30 g/100 g sample db.), and wheat and rice were similar, but almost three times higher, with values around 1.10 g/100 g sample db. (Figure 4). It is well known that the complexes formed by phytic acid are not stable under thermal treatments, and their degradation is also influenced by changes in pH and high pressures, which affect the solubility of phytic acid-cation complexes [31]. However, the lack of effect in purees made from WG and WGF was probably due to the limited water content during autoclaving and the relatively low temperature during cooking (100 °C) [32].

In contrast, infant food from FS samples showed a significant decrease in phytic acid content: in wheat to 0.83 g/100 g purée (−24%), in rice to 0.82 g/100 g puree (−27%), and in maize to 0.15 g/100 g purée (−50%) (Figure 4), since the water excess allows a better dephytinization [11,32].

### 3.5. In Vitro Digestion of Infant Purees

The in vitro starch digestibility of purées made from autoclaved cereals is presented in Table 2, and Figure 5 shows the hydrolyzed starch digested during each phase of in vitro digestion.

A reduction in the total hydrolyzed starch was observed in purée made from WG wheat and WG maize cereals compared to each control (*p* ≤ 0.05) (Table 2). As a result, a significant increase in RS was observed in WG maize purée (17%), in agreement with Liu [33]. The decrease in total hydrolyzed starch (THS), due to a decrease in starch hydrolysis during oral and gastric phases, probably indicates structural rearrangements within the food matrix, as suggested by RVA results. These rearrangements may involve the formation of starch–lipid or starch–lipid–protein complexes during thermal processing [34]. In contrast, the purée prepared from WG rice samples exhibited a higher amount of total hydrolyzed starch with respect to the control, as shown in Table 2. The discrepancies in the results can be attributed to the size of rice granules’ starch (Figure 1), which probably influenced the impact of autoclaving and, thus, the accessibility of digestive enzymes, affecting starch digestibility. As a result, a significant decrease in RS (−11%) and a significant increase in gastric hydrolyzed starch were found in WG rice puree (Figure 5), where structural modifications might break the agglomerates and so increase α-amylase susceptibility of rice starch [35]. The variations in the results can be attributed to the different interactions generated by thermal treatments between the amylose and amylopectin chains of starch, as well as with other components such as lipids and proteins [36]. As a result, these interactions significantly affect the accessibility of digestive enzymes to starch, influencing its digestibility.

Total hydrolyzed starch in purées made from WGF cereals showed an increase of 29%, 70%, and 92% for wheat, rice, and maize, respectively (*p* ≤ 0.05). Similarly, the results for purées made from FS samples showed a significant increase in total hydrolyzed starch, compared to control samples. The enhanced starch hydrolysis observed in WGF purees and FS purees is associated with an elevated degree of starch hydrolysis during the oral phase, compared with controls. These could be explained due to milling of WG before autoclave treatment, which disrupts grain structure, improving enzyme access and starch hydrolysis [37]. Additionally, autoclaving flour suspensions facilitates gelatinization, as indicated by pasting and thermal properties. This process enhances the accessibility of starch in purées made from these samples, resulting in increased susceptibility to enzymatic hydrolysis initially [38]. Additionally, the reduction in phytic acid, as illustrated in Figure 4, contributed to enhanced enzymatic hydrolysis because phytic acid can interfere with starch digestion by inhibiting digestive enzymes and forming complexes with minerals [39].

These results reveal that autoclaving helps the modulation of starch digestibility, which affects health due to the postprandial blood glucose response to carbohydrates [40]. Infant foods characterized by rapid glucose release are associated with digestion predominantly in the oral, gastric, and initial intestinal phases [41], a characteristic that could be necessary in malnutrition, in order to provide a rapid source of energy. Conversely, resistant starch, which is fermented in the colon and produces beneficial short-chain fatty acids, promotes a better development of the infant microbiome [42]. A comprehensive understanding of starch digestion across these phases is essential for optimizing infant food nutritional profiles and developing dietary strategies for metabolic disease prevention and management [6]. Notably, infants require tailored nutritional approaches based on their specific needs [43]. While high resistant starch content promotes intestinal health, infants with malnutrition may benefit from increased oral and gastric starch hydrolysis for rapid energy, as observed in whole grain flour (WGF) purees, or infants who are overweight may require increased intestinal starch hydrolysis for enhanced satiety, as demonstrated by WG maize puree results.

## 4. Conclusions

A nutritionally balanced complementary food without risking safety is essential during the weaning period. While autoclaving remains a prevalent method for sterilizing infant foods, a comprehensive understanding of its impact on nutritional properties is still developing. This study demonstrates that autoclave treatment applied to various cereals with different pretreatments (whole grains, flours, and flour-water suspensions) significantly modifies the physicochemical properties and in vitro digestibility of starch. These modifications include alterations in viscosity, thermal, and mechanical stability due to structural transformations in starch granules, leading to variations in hydrolyzed starch content.

Infant purees made from autoclaved whole grains show a reduction in total hydrolyzed starch and an increase in resistant starch, except for rice, which exhibits increased total starch hydrolysis. Conversely, autoclaved flour (WGF) enhances hydrolyzed starch concentration and decreases resistant starch. Notably, purées made from whole grain flour samples demonstrate an increase in orally digested starch. In autoclaved flour–water suspensions, all samples exhibit complete gelatinization and retrogradation. This process is accompanied by an increase in total hydrolyzed starch and a decrease in phytic acid content. These findings indicate that both the type of matrix used for thermal treatment and the botanical origin of the cereal significantly influence starch properties. The implications for the food industry are substantial, as the incorporation of pretreatments into raw materials for infant food production could effectively modulate starch digestibility and glycemic index during the complementary feeding period. Further research on the feasibility, cost-effectiveness, and scalability of autoclaving methods is crucial for industrial applications.

## Figures and Tables

**Figure 1 foods-14-01367-f001:**
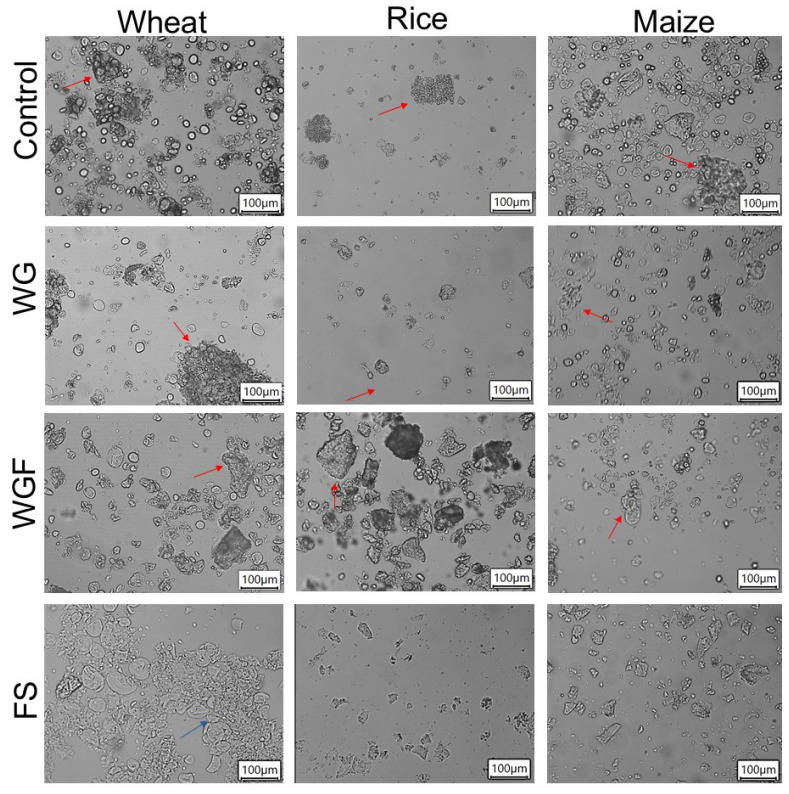
Micrographs of autoclaved cereals observed under a light field at 20× magnification. WG: autoclaved whole grain, WGF: autoclaved whole grain flours, FS: autoclaved flour suspension. Red arrows: amyloplast with embedded starch, blue arrow: empty amyloplast.

**Figure 2 foods-14-01367-f002:**
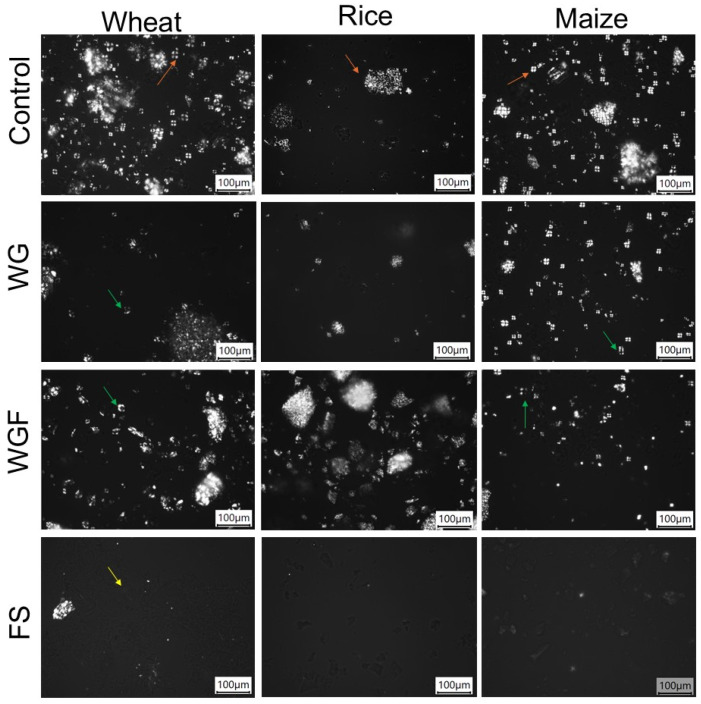
Micrographs of autoclaved cereals observed under polarized-light field at 20× magnification. WG: autoclaved whole grain, WGF: autoclaved whole grain flours, FS: autoclaved flour suspension. Orange arrows: native starch birefringence, green arrows: modified starch birefringence, and yellow arrow: complete loss of starch birefringence.

**Figure 3 foods-14-01367-f003:**
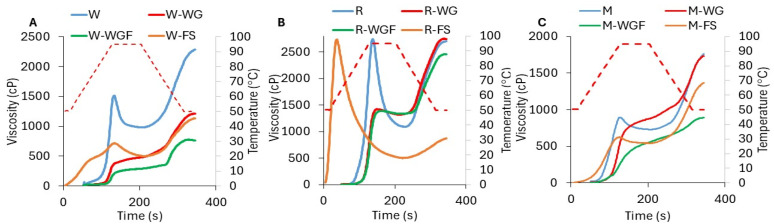
Viscosity profiles of autoclaved cereal samples. (**A**): Wheat; (**B**): rice; and (**C**): maize. Control samples: blue lines; WG: red lines; WGF: green lines; and FS: orange lines. Temperature: dotted red lines.

**Figure 4 foods-14-01367-f004:**
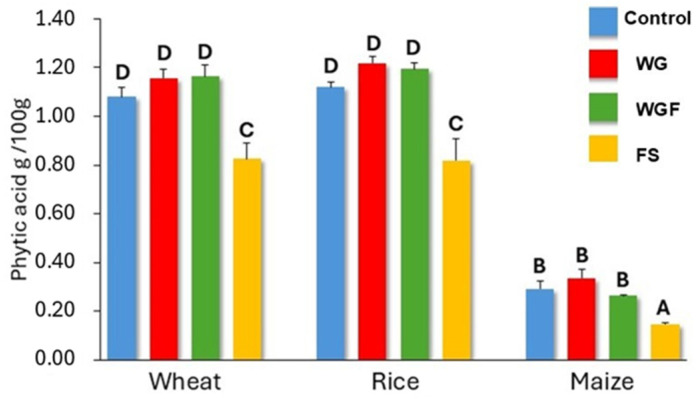
Phytic acid content (grams of PAP (available phosphorus phytate) per 100 g sample db.). Control samples: blue bars; WG: red bars; WGF: green bars, and FS: orange bars. Control: untreated whole grains, WG: autoclaved whole grains, WGF: autoclaved whole grain flour, FS: autoclaved flour suspension. Bars with different letters are significantly different (*p* ≤ 0.05).

**Figure 5 foods-14-01367-f005:**
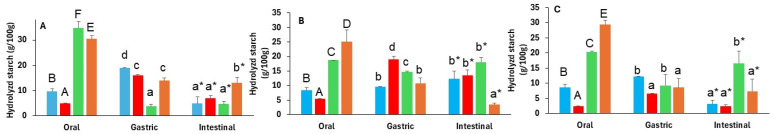
Hydrolyzed starch during the in vitro digestion phases of infant purée. (**A**): Wheat; (**B**): rice; and (**C**): maize. Control samples: blue bars; WG: red bars; WGF: green bars; and FS: orange bars. The three different statistical analyses performed are denoted in capital letters for the oral phase, lower letters for the gastric phase, and a letter with * for the intestinal phase. Different letters mean significant differences (*p* ≤ 0.05).

**Table 1 foods-14-01367-t001:** Thermal properties of control and autoclaved cereal samples.

Samples	Tp (°C)	Temperature Range (°C)	ΔH (J/g)
W	65.2 ± 0.4 ^b^	11.0 ± 0.2 ^a^	8.6 ± 0.4 ^c^
W-WG	64.8 ± 0.1 ^b^	13.1 ± 0.0 ^a^	7.2 ± 0.5 ^b^
W-WGF	67.9 ± 0.2 ^c^	11.1 ± 0.0 ^a^	10.4 ± 1.3 ^d^
W-FS	51.8 ± 0.3 ^a^	12.1 ± 1.8 ^b^	3.1 ± 0.4 ^a^
R	77.5 ± 0.1 ^e^	22.9 ± 2.3 ^b^	8.7 ± 0.1 ^c^
R-WG	78.5 ± 0.0 ^e^	21.3 ± 4.5 ^b^	10.1 ± 0.1 ^d^
R-WGF	78.5 ± 0.4 ^e^	21.8 ± 3.9 ^b^	13.2 ± 0.8 ^e^
R-FS	51.9 ± 0.1 ^a^	14.1 ± 0.9 ^a^	6.7 ± 0.0 ^b^
M	71.0 ± 2.6 ^d^	21.4 ± 2.1 ^b^	7.2 ± 0.7 ^b^
M-WG	72.1 ± 0.0 ^d^	17.3 ± 0.9 ^a^	7.2 ± 0.4 ^b^
M-WGF	71.6 ± 0.7 ^d^	16.3 ± 1.9 ^a^	10.1 ± 0.8 ^d^
M-FS	52.3 ± 0.6 ^a^	14.1 ± 1.2 ^a^	5.6 ± 0.0 ^b^

Values followed by different letters in the same column are significantly different (*p* ≤ 0.05). W: control wheat, R: control rice, M: control maize. WG: autoclaved whole grains, WGF: autoclaved whole grain flour, FS: autoclaved flour suspension.

**Table 2 foods-14-01367-t002:** Effect of thermal treatment on in vitro starch digestion of infant purees.

Samples Used to Made Infant Food	Total Starch Hydrolyzed (g/100 g Starch)	RS (g/100 g Starch)
W	33.3 ± 1.6 c	66.7 ± 1.6 e
W-WG	27.8 ± 0.7 c	72.2 ± 0.7 e
W-WGF	43.1 ± 3.1 e	56.9 ± 3.1 c
W-FS	57.4 ± 2.1 g	42.6 ± 2.1 a
R	30.3 ± 1.8 c	69.8 ± 1.8 e
R-WG	38.1 ± 3.0 d	61.9 ± 3.0 d
R-WGF	51.6 ± 1.9 f	48.4 ± 1.9 b
R-FS	39.3 ± 2.9 d	60.7 ± 2.9 d
M	24.1± 0.2 b	75.9 ± 0.2 f
M-WG	11.3 ± 0.2 a	88.7 ± 0.2 g
M-WGF	46.2 ± 0.6 e	53.8 ± 0.6 c
M-FS	45.4 ± 0.3 e	54.6 ± 0.3 c

Values followed by different letters in the same column are significantly different (*p* ≤ 0.05). W: control wheat, R: control rice, M: control maize. WG: autoclaved whole grains, WGF: autoclaved whole grain flour, FS: autoclaved flour suspension.

## Data Availability

The raw data supporting the conclusions of this article will be made available by the authors on request.

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
