# Peer review of "Influence of Thermal Processing on In Vitro Starch Digestibility in Cereal-Based Infant Foods†"

_foods, 2025, doi:10.3390/foods14081367_

Round 1

Reviewer 1 Report

Comments and Suggestions for Authors

The manuscript investigates the effects of autoclaving (thermal processing at 121°C for 30 minutes) on the starch properties and in vitro digestibility of cereal-based infant purees made from wheat, rice, and maize. The study explores three sample types—whole grains (WG), whole grain flours (WGF), and flour suspensions (FS)—and compares them to untreated controls. The research is timely and relevant, given the importance of optimizing infant nutrition during the complementary feeding period (6–24 months). The use of in vitro digestion models tailored to infant gastrointestinal conditions adds practical value. However, there are areas where clarity, methodological detail, and interpretation could be improved to strengthen the manuscript:

1. Introduction

  • Clarity and Focus: The introduction provides a good overview of complementary feeding and thermal sterilization but could be more concise. The rationale for focusing on autoclaving at 121°C for 30 minutes is not explicitly justified—why this specific condition over others? A sentence linking this to common industrial or home practices would strengthen the motivation.
  • Literature Gaps: While the authors note a lack of studies on conventional autoclaving of whole grains/flours for infant foods, they could better highlight how their work builds on or differs from existing research (e.g., Refs [8, 14, 15]). A brief comparison of autoclaving vs. other thermal methods (e.g., heat-moisture treatment) could sharpen the novelty claim.
  • Suggestion: Add a specific hypothesis or research question to guide the reader (e.g., “We hypothesized that autoclaving would differentially affect starch digestibility based on cereal type and matrix form”).                                                   

    2. Materials and Methods

    • Detail and Reproducibility:
      • The preparation of samples (Section 2.2.1) lacks detail on autoclaving conditions beyond temperature and time (e.g., pressure, vessel type, sample volume). This limits reproducibility.
      • Cooking times for infant food preparation (Section 2.2.6) vary (30 min for wheat/maize, 45 min for rice, 20 min for WGF/FS), but no justification is provided. Were these optimized, or based on texture alone? This could influence digestibility outcomes.
      • The in vitro digestion protocol (Section 2.2.8) references Rodriguez [17], but key parameters (e.g., incubation times, sample amounts) should be briefly summarized here for clarity.
    • Statistical Analysis: The use of ANOVA and DGC tests is appropriate, but the manuscript does not specify the number of replicates beyond “duplicates” for DSC (Section 2.2.4). Were all other analyses (e.g., RVA, digestion) also in duplicate, or more? This affects statistical power.
    • Suggestion: Provide more procedural details (e.g., autoclave pressure, cooking rationale) and clarify replication across all experiments.

    3. Results and Discussion

    • Data Presentation:
      • The results are well-supported with figures and tables, but some figures (e.g., Figs 1, 2) are referenced without being included in the provided text, making it hard to assess their quality or relevance.
      • Table 1 and Table 2 are clear, but the text often repeats numerical data verbatim (e.g., THS values in Section 3.5). This could be streamlined by referring to tables more effectively.
    • Interpretation:
      • The cereal-specific differences (e.g., rice vs. wheat/maize in WG samples) are intriguing but underexplored. The authors attribute rice’s higher THS to smaller granule size (Section 3.5), but this is speculative without supporting granule size measurements or citations beyond [31]. More evidence or discussion of starch composition (e.g., amylose/amylopectin ratios) could strengthen this.
      • The discussion of phytic acid reduction (Section 3.4) is insightful, but the lack of effect in WG/WGF vs. FS is dismissed as due to “limited water content” without quantifying moisture levels or citing supporting studies beyond [28]. This weakens the argument.
      • The link between viscosity changes (Section 3.2) and digestibility (Section 3.5) is not explicitly drawn, despite both being affected by starch structure. Integrating these findings could enhance the narrative.

    4. Conclusions

    • Scope and Impact: The conclusions summarize key findings well but overstate the implications slightly by suggesting “substantial” food industry applications without discussing feasibility (e.g., cost, scalability of autoclaving flour suspensions). A more tempered statement would be appropriate.

Comments on the Quality of English Language

Some improvements are needed. 

Author Response

Response to Reviewer 1 Comments

1. Summary

Thank you for taking the time to review this manuscript. Detailed responses are provided below, and corrections are highlighted in red in the re-submitted files.

3. Point-by-point response to Comments and Suggestions for Authors

Comments 1: The introduction provides a good overview of complementary feeding and thermal sterilization but could be more concise. The rationale for focusing on autoclaving at 121°C for 30 minutes is not explicitly justified—why this specific condition over others? A sentence linking this to common industrial or home practices would strengthen the motivation

Response 1: Thank you for pointing this out. We agree with this comment. Therefore, we have added a paragraph according to your suggestion. [54-60 lines.]

Comments 2: Literature Gaps: While the authors note a lack of studies on conventional autoclaving of whole grains/flours for infant foods, they could better highlight how their work builds on or differs from existing research (e.g., Refs [8, 14, 15]). A brief comparison of autoclaving vs. other thermal methods (e.g., heat-moisture treatment) could sharpen the novelty claim.

Response 2:

Thank you for this insightful comment regarding the literature gaps. We agree that a clearer connection to existing research and a brief comparison with other thermal methods would strengthen the manuscript's novelty claim. Following your suggestion, we have added a paragraph in the Introduction to explicitly highlight how our work builds upon and differs from previous studies. The paragraph was inserted between lines 38 and 50.

Comments 3:
Suggestion
: Add a specific hypothesis or research question to guide the reader (e.g., “We hypothesized that autoclaving would differentially affect starch digestibility based on cereal type and matrix form”). 

Response 3:

A clear hypothesis has been added to guide the reader. You can find it on lines 86-89. Thank you for the suggestion.

Comments 4:
Detail and Reproducibility:

The preparation of samples (Section 2.2.1) lacks detail on autoclaving conditions beyond temperature and time (e.g., pressure, vessel type, sample volume). This limits reproducibility. 

Response 4:

Acknowledged. We've added specific autoclaving details (pressure, vessel, volume) to the revised Section 2.2.1 to improve reproducibility, 98-105 lines. Thank you 

Comments 5:

Cooking times for infant food preparation (Section 2.2.6) vary (30 min for wheat/maize, 45 min for rice, 20 min for WGF/FS), but no justification is provided. Were these optimized, or based on texture alone? This could influence digestibility outcomes.  

Response 5:

We appreciate your observation regarding the different cooking times outlined in Section 2.2.6 and the need for justification. As referenced in the citation, these times were carefully established to ensure complete starch gelatinization for each sample type. Our method involved extracting grains at specific time points during cooking and visually assessing them between two glass slides to confirm full gelatinization had occurred internally. This focus on complete starch gelatinization was our primary criterion for determining the cooking times. 

Comments 6:

The in vitro digestion protocol (Section 2.2.8) references Rodriguez [17], but key parameters (e.g., incubation times, sample amounts) should be briefly summarized here for clarity.

Response 6:

Thank you for this suggestion. We agree that summarizing the key parameters of the in vitro digestion protocol would improve clarity. We have now added a summary of the incubation times and sample amounts used in Section 2.2.8, 151-164 lines.

Comments 7: Statistical Analysis: The use of ANOVA and DGC tests is appropriate, but the manuscript does not specify the number of replicates beyond “duplicates” for DSC (Section 2.2.4). Were all other analyses (e.g., RVA, digestion) also in duplicate, or more? This affects statistical power.

Response 7:

Thank you for raising this important point about statistical analysis. We have clarified in the revised manuscript, specifically on lines 118-120 and 174-175, that all analyses, including RVA and digestion, were also performed in duplicate to ensure adequate statistical power. 

Comments 8:
Suggestion: Provide more procedural details (e.g., autoclave pressure, cooking rationale) and clarify replication across all experiments.

Response 8:

Thank you for this helpful suggestion. We have now added more procedural details regarding the autoclave pressure and the rationale behind the cooking times in the revised manuscript. The additional data are in lines 139-143 and 102-103.

Comments 9:
Data Presentation:

The results are well-supported with figures and tables, but some figures (e.g., Figs 1, 2) are referenced without being included in the provided text, making it hard to assess their quality or relevance.

Response 9:

Thank you for noting this. We apologize for the omission. Figures 1 and 2 are now included in the revised submission. We hope this helps in assessing our results.

Comments 10:
Table 1 and Table 2 are clear, but the text often repeats numerical data verbatim (e.g., THS values in Section 3.5). This could be streamlined by referring to tables more effectively.

Response 10:

Thank you for this constructive feedback. We agree that the repetition of numerical data can be streamlined. In the revised manuscript, we have made efforts to refer to Tables 1 and 2 more effectively in the text, particularly in Section 3.5, to avoid verbatim repetition and improve readability.

Comments 11:
Interpretation:

The cereal-specific differences (e.g., rice vs. wheat/maize in WG samples) are intriguing but underexplored. The authors attribute rice’s higher THS to smaller granule size (Section 3.5), but this is speculative without supporting granule size measurements or citations beyond [31]. More evidence or discussion of starch composition (e.g., amylose/amylopectin ratios) could strengthen this.

Response 11:

We appreciate your valuable feedback. We have considered your suggestion and integrated additional discussion and evidence pertaining to starch composition and granule size into the revised manuscript, specifically in lines 300-304.

Comments 12:

The discussion of phytic acid reduction (Section 3.4) is insightful, but the lack of effect in WG/WGF vs. FS is dismissed as due to “limited water content” without quantifying moisture levels or citing supporting studies beyond [28]. This weakens the argument.

Response 12:

Feedback noted regarding the "limited water content" explanation. Additional citation on water content's role in phytic acid reduction during thermal processing has been included in revised Section 3.4, line 277.

Comments 13: The link between viscosity changes (Section 3.2) and digestibility (Section 3.5) is not explicitly drawn, despite both being affected by starch structure. Integrating these findings could enhance the narrative.

Response 13:

Thank you for your suggestion. We agree that explicitly linking the observed viscosity changes with the digestibility results would strengthen the narrative. We performed a correlation analysis but found no direct correlations between the specific values. Instead, the relationship appears to be with the overall behavior rather than between individual variables (lines: [291 y 318]).

Comments 14:
Scope and Impact: The conclusions summarize key findings well but overstate the implications slightly by suggesting “substantial” food industry applications without discussing feasibility (e.g., cost, scalability of autoclaving flour suspensions). A more tempered statement would be appropriate.

Response 14:

We agree that a moderate statement on industrial applications is necessary. The conclusion now reflects the need for further research into feasibility, cost, and scalability for food industry applications, lines 361-363.

Reviewer 2 Report

Comments and Suggestions for Authors

This study used thermal treatment to modulate starch digestibility and viscosity properties in whole grains, whole grain flours, and whole grain flour/water suspension, and plan to optimize infant nutrition foods. This is a very interesting study, and will promote the infant food industry. The paper fits with Foods.

  • The authors did not add row number, which is better to review;
  • In the methods, the authors gave the autoclaving treatment 30 min, why not to show this treatment time in abstract and conclusion section. The authors had better have data to demonstrate this best treatment time.
  • The authors used whole grains (WG), whole grain flours (WGF), whole grain flour/water suspension (FS). It is better to use these names in the whole text. In some places, the “”flour” is not easy to understand. Moreover, the particle size in whole grain flours (WGF) should be given. When preparing infant puree, the particle size of purees from these three materials should be given.
  • In the last paragragh of introduction, the scientific hypothesis and innovation points should be given.
  • In figures 1-2, the gap among plates should be a little bigger.
  • For pasting properties of samples, there is a table to show the data of different samples.
  • In table 1, the conclusion temperature of DSC gelatinization should be given, this can reflect the starch retrogradation.
  • In table 2, RS should be noted.
  • In the abstract and conclusion section, the authors did not mention phytic acid content.
  • As the whole cereal grains as infant foods, the ash content and taste question should be considered in this paper.

Author Response

Response to Reviewer 2 Comments

1. Summary

Thank you for taking the time to review this manuscript. Detailed responses are provided below, and corrections are highlighted in red in the re-submitted files.

3. Point-by-point response to Comments and Suggestions for Authors

Comments 1: In the methods, the authors gave the autoclaving treatment 30 min, why not to show this treatment time in abstract and conclusion section. The authors had better have data to demonstrate this best treatment time

Response 1:

Thank you for the feedback. The 30-minute autoclaving time is justified in the Introduction section on lines [56-60]. Our study shows the effects of this common sterilization time.

Comments 2: The authors used whole grains (WG), whole grain flours (WGF), whole grain flour/water suspension (FS). It is better to use these names in the whole text. In some places, the ”flour” is not easy to understand. Moreover, the particle size in whole grain flours (WGF) should be given. When preparing infant puree, the particle size of purees from these three materials should be given.

Response 2:

Thank you for the suggestions. We will consistently use WG, WGF, and FS. We will also add particle size data for WGF and purees if available, or clarify if not measured, line 100.

Comments 3:
Suggestion
: In the last paragraph of introduction, the scientific hypothesis and innovation points should be given

Response 3:

A clear hypothesis has been added to guide the reader. You can find it on lines 86-89. Thank you for the suggestion.

Comments 4:
In figures 1-2, the gap among plates should be a little bigger.

Response 4:

We have increased the gap between the plates in the revised Figures 1 and 2 as suggested. 

Comments 5:

For pasting properties of samples, there is a table to show the data of different samples.

Response 5:

Pasting properties are presented in Figure 3 and some values were highligted in the text. 

Comments 6:

In table 1, the conclusion temperature of DSC gelatinization should be given, this can reflect the starch retrogradation.

Response 6:

Table 1 shows the onset temperature and the range of temperature in which the gelatinization/retrogardation take place, so that including the endset temperature could result in innecesarry information that could leads to misunderstanding

Comments 5:

 In table 2, RS should be noted.

Response 5:

We agree that noting the Resistant Starch (RS) data in Table 2 would be beneficial for clarity and a more comprehensive presentation of the results. We will ensure that RS is clearly indicated in the revised Table 2.

Comments 6:
In the abstract and conclusion section, the authors did not mention phytic acid content.

Response 6:

We agree that the phytic acid content is a relevant finding and should be highlighted in the Abstract and Conclusion sections. We will revise these sections to include a brief mention of the key findings related to phytic acid content, lines 18 y 354-356.

Comments 7:
As the whole cereal grains as infant foods, the ash content and taste question should be considered in this paper.

Response 7:

Characterization of cereals used in this study is part of a previous work that is now included in the manuscript.